# Heterozygous diploid and interspecies SCRaMbLEing

Michael J. Shen [1], Yi Wu [2,3], Kun Yang[4,5], Yunxiang Li[2,3], Hui Xu[2,3], Haoran Zhang[2,3], Bing-Zhi Li[2,3], Xia Li [2,3], Wen-Hai Xiao[2,3], Xiao Zhou[2,3], Leslie A. Mitchell[1], Joel S. Bader [4,5], Yingjin Yuan [2,3] & Jef D. Boeke[1]

SCRaMbLE (Synthetic Chromosome Rearrangement and Modification by LoxP-mediated Evolution) is a genome restructuring technique that can be used in synthetic genomes such as that of Sc2.0, the synthetic yeast genome, which contains hundreds to thousands of strategically positioned loxPsym sites. SCRaMbLE has been used to induce rearrangements in yeast strains harboring one or more synthetic chromosomes, as well as plasmid DNA in vitro and in vivo. Here we describe a collection of heterozygous diploid strains produced by mating haploid semisynthetic Sc2.0 strains to haploid native parental strains. We subsequently demonstrate that such heterozygous diploid strains are more robust to the effects of SCRaMbLE than haploid semisynthetic strains, rapidly improve rationally selected phenotypes in SCRaMbLEd heterozygous diploids, and establish that multiple sets of independent genomic rearrangements are able to lead to similar phenotype enhancements. Finally, we show that heterozygous diploid SCRaMbLE can also be carried out in interspecies hybrid strains.

[1] Department of Biochemistry Molecular Pharmacology and Institute for Systems Genetics, NYU Langone Health, New York, NY 10016, USA. [2] Key Laboratory of Systems Bioengineering (Ministry of Education), School of Chemical Engineering and Technology, Tianjin University, 300072 Tianjin, China. [3] SynBio Research Platform, Collaborative Innovation Center of Chemical Science and Engineering (Tianjin), Tianjin University, 300072 Tianjin, China. [4] Department of Biomedical Engineering, Johns Hopkins University, Baltimore, MD 21218, USA. [5] High Throughput Biology Center, Johns Hopkins University School of Medicine, Baltimore, MD 21205, USA. These authors contributed equally: Michael J. Shen, Yi Wu. Correspondence and requests for materials should be addressed to J.D.B. (email: Jef.Boeke@nyumc.org)

The SCRaMbLE system, developed as part of the Sc2.0 project, enables inducible rearrangement of synthetic chromosomes by the Cre recombinase enzyme. The design of synthetic chromosomes[1] specifies the insertion of the palindromic 34 bp loxPsym[2] recombination site 3 bp downstream of stop codons of all nonessential open reading frames (ORFs). Additional loxPsym sequences are inserted in place of deleted non-intronic features and a thinning algorithm ensures that minimum inter-loxPsym site distance is greater than 300 bp. Conventional loxP sites are directional, and the relative orientation of any pair of loxP sites dictates whether a deletion, inversion, or translocation will occur. Because loxPsym sites are nondirectional[2], they enable the stochastic generation of deletions, duplications, inversions, and/or translocations within and between synthetic chromosomes[3].

The SCRaMbLE system allows for exploration and characterization of a huge number of potential genomic rearrangements via expression of Cre recombinase in the nucleus of synthetic chromosome-bearing cells[4]. Controlling the activity of Cre is important for maintaining Sc2.0 chromosome stability; to implement this, Cre is fused to the estrogen binding domain (EBD)[5] of the estrogen receptor, which effectively sequesters Cre-EBD in the cytosol. Only upon treatment with estradiol does Cre-EBD translocate into the nucleus and become available to recombine loxPsym sites. Cre-EBD can also be regulated at a transcriptional level by cell-cycle specific or constitutive promoters. This system can generate strains with phenotypes that differ from their non-SCRaMbLEd parent.

The random nature of SCRaMbLE events can also lead to a number of sub-optimal outcomes with regards to studying rearrangements in an unbiased fashion. SCRaMbLE of haploid strains bearing one or more synthetic chromosomes results in a high lethality rate due to the deletion of one or more essential genes[3,6]. Additionally, deletion of important but nonessential genes may mask an otherwise apparent change of phenotype. Finally, SCRaMbLE of synthetic chromosome-bearing strains has thus far been carried out in a Saccharomyces cerevisiae laboratory strain background[7], limiting its industrial applications.

Here, we address the above caveats of SCRaMbLE by constructing a set of heterozygous diploid yeast strains. We demonstrate that SCRaMbLE in heterozygous diploids results in a higher proportion of surviving cells in strains bearing both one and two synthetic chromosomes. Subsequently, we perform SCRaMbLE both in an S. cerevisiae and a S. cerevisiae/S. paradoxus heterozygous diploid (i.e., interspecies hybrid) strain to rapidly evolve heat and caffeine tolerance (respectively), and identify genomic rearrangements responsible for the observed phenotypic alterations.

## Results

### Characterization of SCRaMbLE in diverse heterozygous yeast.
Each member of this collection was produced by mating a haploid strain bearing either one (synX) or two (synV and synX) synthetic chromosomes[8,9] with a haploid strain from the Saccharomyces Genome Resequencing Project (SGRP) set[10,11]. SynX is 707,459 base pairs in length and encodes 245 loxPsym sites, while synV is 536,024 base pairs long and carries 176 loxPsym sites. The SGRP set contains both S. cerevisiae and S. paradoxus haploids, both of which were successfully mated to synX and synV-synX strains to generate two series of intraspecies and interspecies heterozygous diploids (Fig. 1a, Supplementary Table 1).

Upon generation and validation of the heterozygous diploid strains, we first demonstrated their increased tolerance to the effects of SCRaMbLE. Strains were transformed with an episomal, URA3-bearing plasmid encoding Cre-EBD. We chose to use

Cre-EBD driven by the G2/M CLB2 promoter (pCLB2) in contrast to previous studies that used the daughter-specific SCW11 promoter (pSCW11) for several reasons. First, pCLB2 provides a pulse of Cre-EBD expression once per cell cycle[12] as compared to pSCW11, which is activated only in newborn (daughter) cells. Thus, not only should every cell in the population be affected by estradiol exposure rather than only newborn cells, but repeated exposure to Cre-EBD should theoretically yield more recombination events. Second, activation in G2/M when the genome is present in two copies may yield an increase in the number of SCRaMbLE-mediated duplication events. Third, we found that the pCLB2-Cre-EBD construct was less toxic in a haploid context to synthetic chromosome-bearing strains in the absence of estradiol as compared to pSCW11-Cre-EBD (Supplementary Fig. 1), perhaps because CLB2 expression is G2-specific. Following SCRaMbLE induction for 6 h with 1 μM β-estradiol, cells were washed, diluted and plated on yeast extract peptone dextrose (YPD) agar plates for assessment of SCRaMbLE-induced lethality. We found that all heterozygous diploid strains tested were substantially more tolerant of Cre-mediated SCRaMbLE compared to haploid strains (Fig. 1b, c). Survival of haploid semisynthetic strains subjected to SCRaMbLE was generally less than 30% when compared to non-SCRaMbLEd parents, while survival of heterozygous diploid strains was generally upwards of 70%. Careful inspection of the plated cells revealed an increase in the frequency of slow-growing colonies that arise upon estradiol pretreatment; such small colonies typically show rearrangements when they arise in semisynthetic haploid strains, suggesting that SCRaMbLE is functioning in the wide range of strain backgrounds. We further confirmed that heterozygous diploid strains with two synthetic chromosomes were capable of undergoing SCRaMbLE while displaying less recombinase-mediated cell death compared to haploid strains (Fig. 1b). We did not notice any difference in behavior between heterozygous diploid strains constructed with S. cerevisiae strains from the SGRP and those constructed with S. paradoxus strains, suggesting that SCRaMbLE of heterozygous diploids can be applied to Saccharomyces interspecies strain combinations.

### SCRaMbLE improves thermotolerance in a Y12-synX diploid.
We next sought to determine whether SCRaMbLE could improve the ability of heterozygous diploid strains to tolerate extreme drug or environmental conditions. We examined a heterozygous diploid strain composed of the Y12 sake-brewing S. cerevisiae strain mated with a synX-bearing strain (Y12-synX). Y12 was chosen for its relative thermotolerance compared to other S. cerevisiae strains, as well as its use in an industrially relevant process. Y12-synX cells were subjected to SCRaMbLE as described above and selected at 42 °C. Thermotolerant single colonies were grown in liquid YPD cultures with daily serial dilution for one week to ensure loss of the Cre plasmid and then grown at 30 °C, 37 °C, 40 °C, and 42 °C. Multiple independent SCRaMbLEd isolates displayed an improvement in growth at 42 °C when compared to parent non-SCRaMbLEd strains (Fig. 2a). Interestingly, isolate yYW166 grew well at 42 °C but displayed decreased fitness at both 30 °C and 37 °C compared to its non-SCRaMbLEd parent strain while isolate yYW167 grew well at all temperatures tested. At high temperatures, one round of SCRaMbLE was sufficient to recover growth of yYW167 to the level of the Y12 homozygous diploid.

We used whole genome sequencing to examine the recombination events that occurred in yYW166 and yYW167 compared to their parent strain, yMS423. Because SCRaMbLE events by nature act on the stretches of DNA between loxPsym sites as discrete units, we decomposed synX into 248 segments, with each

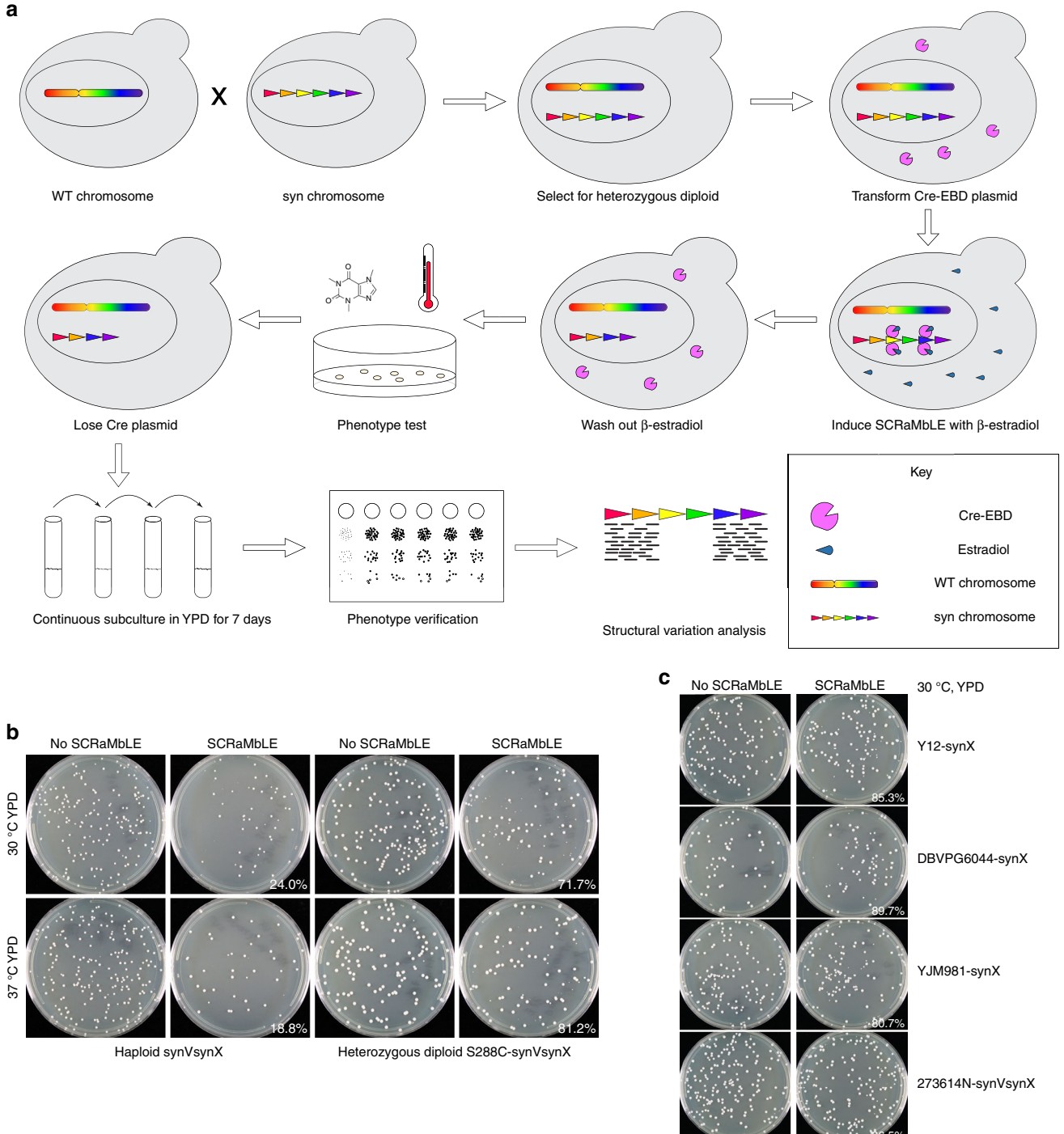

**Fig. 1** Construction and testing of heterozygous diploid strains. **a** Experimental workflow. A *S. cerevisiae* strain bearing one or more synthetic chromosomes is mated to a *S. cerevisiae* or *S. paradoxus* strain with a "wild-type" genome. The resultant heterozygous diploid cells can be selected for, SCRaMbLEd, and tested for tolerance of a variety of environmental and chemical conditions. Strains showing increased fitness have their phenotype verified and can be analyzed with whole genome sequencing to determine the sets of SCRaMbLE events responsible for a given phenotype. **b** SCRaMbLE of haploid and heterozygous diploid synVsynX yeast was induced by adding 1 μM β-estradiol to culture media for 6 h. Heterozygous diploid S288C-synVsynX strains demonstrate a lesser degree of SCRaMbLE-mediated lethality at both 30 °C and 37 °C compared to haploid synVsynX strains. **c** Heterozygous diploid strains incorporating a variety of *S. cerevisiae* "wild-type" genomes are robust to SCRaMbLE. Additionally, heterozygous diploid strains containing two synthetic chromosomes can be SCRaMbLEd without appreciable loss in viability compared to strains containing one synthetic chromosome

segment flanked by one (in the case of the first and last segments) or two loxPsym sites. By determining the emergence of novel junctions between non-adjacent segments and using average coverage across a segment to calculate its copy number, we were able to infer some of the structural changes caused by

SCRaMbLE. Our analysis showed that yYW166 had four deletions encompassing the ORFs YJL222W-YJL217W, YJL161W-YJL130C, YJL052C-YJL028W, and YJR093C-YJR159W, a smaller deletion of an intergenic sequence, and a duplication in the YJL027C-YJL022W region (Fig. 2b,

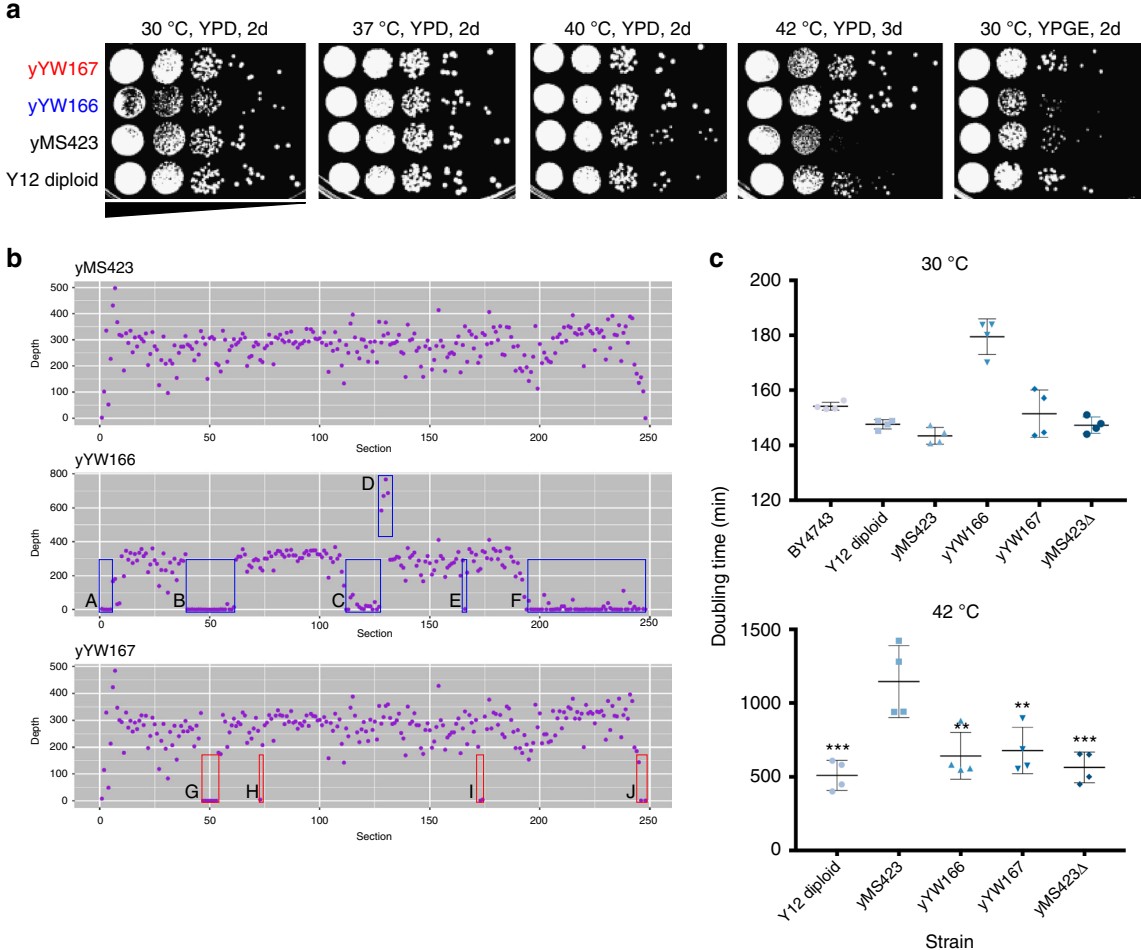

**Fig. 2** SCRaMbLE of Y12-synX rapidly improves thermotolerance. **a** Serial dilution assay comparing the growth of two independent SCRaMbLEd isolates of Y12-synX (yYW166 and yYW167) with the non-SCRaMbLEd parent strain (yMS423) and a Y12 diploid strain (yYW207) under various temperature conditions, as well as in YPGE. **b** Average sequencing depth per segment along synX of yMS423, yYW166, and yYW167. Deletions (boxes A, B, C, E, F, G, H, I, and J), as well as a duplication (box D) are highlighted. **c** BY4743, a Y12/Y12 diploid, yMS423, yYW166, yYW167, and yMS423Δ were all grown in liquid YPD overnight and diluted to a starting $A_{600}$ of 0.1 in fresh YPD. These strains were then cultured in a 96-well plate reader with shaking at 30 °C or 42 °C. Optical density measurements were taken every 10 min and used to calculate doubling time. Error bars shown are mean and standard deviation from four technical replicates. One-way ANOVA with multiple comparisons was used to assess difference between each sample and yMS423 (***$p < 0.001$, **$p < 0.01$). Variance between the groups was determined to be similar

Supplementary Fig. 2). yYW167 had two deletions spanning the ORFs YJL154C-YJL140W and YJR055W-YJR056C along with two smaller deletions. Deletion of the YJL154C-YJL140W region, spanning 20,815 bp, was common to both strains. This region includes *TIM17* (YJL143W), an essential gene[13] whose deletion would have been lethal had this SCRaMbLE event occurred in a semisynthetic haploid cell. We employed CRISPR-Cas9 to delete the region spanning YJL154C-YJL140W in yMS423, using a guide RNA targeted to a synX sequence not predicted to be found in Y12, as well as a donor DNA with homology upstream of YJL154C and downstream of YJL140W. This strain (yMS423Δ) grew equally well compared to yMS423 at 30 °C, but was better able to tolerate higher temperature (42 °C), similar to yYW166 and yYW167 (Fig. 2c, Supplementary Fig. 3). These data provide evidence that multiple independent rearrangements can result in similar phenotypes in SCRaMbLEd heterozygous diploids and that SCRaMbLE of heterozygous diploids can recover genotypes that cannot be found by SCRaMbLE of semisynthetic haploids.

**Interspecies SCRaMbLE links *POL32* to caffeine tolerance**. To test the interspecies SCRaMbLE system, we chose a candidate obtained by mating the *S. paradoxus* strain CBS5829 with a synX bearing strain (CBS5829-synX) based on the robust relative tolerance of CBS5829 to caffeine[10]. Thus, the parental strain here is actually an interspecies hybrid. Caffeine, like the macrolide antibiotic rapamycin, is an inhibitor of the TOR kinase cascade in both budding yeast and the fission yeast *Schizosaccharomyces pombe*, leading to increased chronological lifespan[14,15]. Single, 6-h exposures of CBS5829-synX cells to β-estradiol were used to induce SCRaMbLE. We detected differential growth between SCRaMbLEd and non-SCRaMbLEd cells in liquid YPD supplemented with 5 and 7 mg/mL caffeine, a phenotype that was subsequently verified on solid medium with serial dilution assays (Fig. 3a). Strains exhibiting caffeine-tolerant phenotypes were obtained in multiple independent SCRaMbLE experiments with selection immediately post-SCRaMbLE on solid medium containing caffeine or in liquid YPD containing caffeine.

Whole genome sequencing was used to examine the changes caused by SCRaMbLE in 10 independent isolates of caffeine tolerant heterozygous diploids. We were able to detect each of deletions, inversions, and duplications in SCRaMbLEd strains, with no two isolates having identical sets of chromosomal rearrangements. However, we did recover the same duplication of

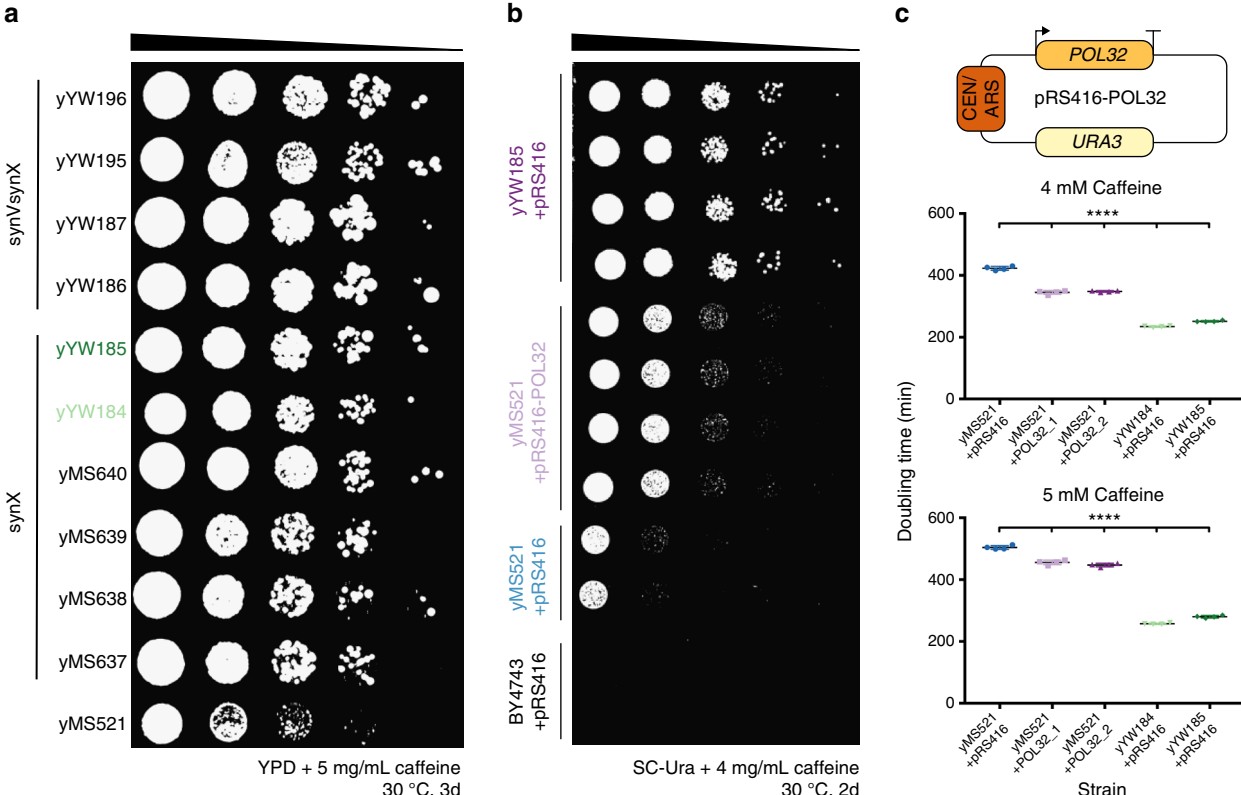

**Fig. 3** SCRaMbLE of CBS5829-syn(V)X improves caffeine tolerance. **a** Serial dilution assay comparing the growth of SCRaMbLEd *S. paradoxus* CBS5829-synX (yMS637, yMS638, yMS639, yMS640, yYW184, and yYW185) or CBS5829-synVsynX (yYW186, yYW187, yYW195, and yYW196) strains to their non-SCRaMbLEd CBS5829-synX parent (yMS521) on high caffeine YPD plates. **b** The *POL32* gene with 500 bp upstream/300 bp downstream sequence was cloned from BY4741 into the episomal plasmid pRS416 and the resulting plasmid pRS416-*POL32* transformed into yMS521. These strains were compared via serial dilution assay to the SCRaMbLEd strain yYW185 transformed with pRS416, yMS521 transformed with pRS416, or BY4743 transformed with pRS416 on SC–Ura + 4 mg/mL caffeine. **c** yMS521 transformed with pRS416, yMS521 transformed with pRS416-*POL32*, yYW184 transformed with pRS416, and yYW185 transformed with pRS416 were all grown in liquid SC-Ura media overnight and then diluted to a starting $A_{600}$ of 0.1 in either SC–Ura + 4 mg/mL or 5 mg/mL caffeine and cultured in a 96-well plate reader with shaking. Optical density measurements were taken every 10 min and used to calculate doubling time. Error bars shown are mean and standard deviation from four technical replicates. One-way ANOVA with multiple comparisons was used to assess difference between yMS521 + pRS416 and other samples (****$p$ < 0.0001). Variance between the groups was determined to be similar

synX segments in SCRaMbLE derivatives from two independent experiments (Supplementary Fig. 4). The duplications in the two strains were molecularly distinct, with one strain containing 7 adjacent upstream segments duplicated, and another containing one adjacent downstream segment duplicated. The synX duplication of segments 163 and 164 encompasses *POL32*, a nonessential subunit of DNA polymerase δ[16]. *POL32* has been associated with roles in DNA damage repair and chromosomal DNA replication, and previous work examining *POL32* null mutants demonstrated an increased resistance to rapamycin[17]. Interestingly, and consistent with earlier work, SCRaMbLEd strains with *POL32* duplications (yMS637 and yYW185) were less tolerant of rapamycin than the non-SCRaMbLEd parental strain yMS521 (Supplementary Figs. 5, 6). Introducing an episomal copy of *POL32* under its native promoter to the non-SCRaMbLEd CBS5829-synX strain was sufficient to increase its caffeine tolerance on solid media (Fig. 3b), although not to levels as high as SCRaMbLEd strains. We additionally quantified doubling time in caffeine-containing synthetic complete liquid media and observed a similar trend (Fig. 3c). To ensure that SNVs were not driving caffeine tolerance in yMS637 and yYW185, we performed variant calling analysis. We removed from consideration variants common to yMS637, yYW185, and parental non-SCRaMbLEd strain yMS521. We then examined variants shared by both yMS637 and yMS185 and found no evidence for

causative variants (Supplementary Table 2). This result points to the utility of heterozygous diploid SCRaMbLE in identifying genes previously unassociated with particular drug tolerances.

We extended both phenotype testing and WGS analysis to a heterozygous diploid CBS5829-synVsynX containing two synthetic chromosomes. SCRaMbLE of heterozygous diploids with two synthetic chromosomes also proved more robust than their haploid counterparts (Fig. 1c). SCRaMbLE also increased the caffeine tolerance of CBS5829-synVsynX strains (Fig. 3a). From the 4 strains sequenced, we did not observe conserved rearrangements in SCRaMbLEd CBS5829-synVsynX strains or rearrangements common to both CBS5829-synX and CBS5829-synVsynX strains.

For experiments selecting for Y12-synX and CBS5829-synX strains on heat and caffeine, respectively, we also SCRaMbLEd an S288C-synX strain and selected the resulting cells along with our strains of interest. In neither case did we observe SCRaMbLEd colonies that grew at 42 °C (Supplementary Fig. 7) or on media with 5 (or 7) mg/mL caffeine.

## Discussion

In this work, we have described a set of strains and methodology that provide improvements to the Sc2.0 SCRaMbLE system. By inducing SCRaMbLE in heterozygous diploids that contain a wild-type counterpart to each synthetic chromosome, we reduce

**Table 1 Strains used in this study**

| Strain name | Description | Genotype |
|---|---|---|
| BY4741 | | MATa his3Δ1 leu2Δ0 met15Δ0 ura3Δ0 |
| BY4743 | | MATa/α his3Δ1/his3Δ1 leu2Δ0/leu2Δ0 LYS2/lys2Δ0 met15Δ0/ MET15 ura3Δ0/ura3Δ0 |
| yYW117 | BY4741 containing synX | MATa his3Δ1 leu2Δ0 met15Δ0 ura3Δ0 SYN10 ho::tR(ccu)J |
| yYW168 | synX lys2::NatMX | MATa his3Δ1 leu2Δ0 met15Δ0 ura3Δ0 SYN10 ho::tR(ccu)J lys2:: NatMX |
| yYW139 | BY4741 containing synVsynX | MATa his3Δ1 leu2Δ0 met15Δ0 ura3Δ0 SYN5 SYN10 ho::tR(ccu)J |
| yYW169 | synVX lys2::NatMX | MATa his3Δ1 leu2Δ0 met15Δ0 ura3Δ0 SYN5 SYN10 ho::tR(ccu)J lys2:: NatMX |
| yMS253 | S288C alpha haploid | MATα ura3::KanMX ho::Hyg |
| yMS275 | Y12 alpha haploid | MATα ura3::KanMX ho::Hyg |
| yMS354 | CBS5829 alpha haploid | MATα ura3::KanMX ho::Hyg |
| yYW207 | Y12 diploid | MATa/α ura3::KanMX/ura3::KanMX ho::Hyg/ho::Hyg |
| yYW208 | CBS5829 diploid | MATa/α ura3::KanMX/ura3::KanMX ho::Hyg/ho::Hyg |
| yMS401 | Diploid of yYW168 and yMS253 | MATa/α ura3Δ0/ura3::KanMX ho::tR(ccu)J/ho::Hyg lys2::NatMX/ LYS2 SYN10/WT10 |
| yMS426 | Diploid of yYW169 and yMS253 | MATa/α ura3Δ0/ura3::KanMX ho::tR(ccu)J/ho::Hyg lys2::NatMX/ LYS2 SYN5/WT5 SYN10/WT10 |
| yMS423 | Diploid of yYW168 and yMS275 | MATa/α ura3Δ0/ura3::KanMX ho::tR(ccu)J/ho::Hyg lys2::NatMX/ LYS2 SYN10/WT10 |
| yMS423Δ | Diploid of yYW168 and yMS275 with deletion of YJL154C to YJL140W in synX | MATa/α ura3Δ0/ura3::KanMX ho::tR(ccu)J/ho::Hyg lys2::NatMX/ LYS2 SYN10/WT10 |
| yMS448 | Diploid of yYW169 and yMS275 | MATa/α ura3Δ0/ura3::KanMX ho::tR(ccu)J/ho::Hyg lys2::NatMX/ LYS2 SYN5/WT5 SYN10/WT10 |
| yMS521 | Diploid of yYW168 and yMS354 | MATa/α ura3Δ0/ura3::KanMX ho::tR(ccu)J/ho::Hyg lys2::NatMX/ LYS2 SYN10/WT10 |
| yMS548 | Diploid of yYW169 and yMS354 | MATa/α ura3Δ0/ura3::KanMX ho::tR(ccu)J/ho::Hyg lys2::NatMX/ LYS2 SYN5/WT5 SYN10/WT10 |
| yYW166 | yMS423, SCRaMbLEd, heat tolerant | MATa/α ura3Δ0/ura3::KanMX ho::tR(ccu)J/ho::Hyg lys2::NatMX/ LYS2 SYN10/WT10 |
| yYW167 | yMS423, SCRaMbLEd, heat tolerant | MATa/α ura3Δ0/ura3::KanMX ho::tR(ccu)J/ho::Hyg lys2::NatMX/ LYS2 SYN10/WT10 |
| yYW184 | yMS521, SCRaMbLEd, caffeine tolerant | MATa/α ura3Δ0/ura3::KanMX ho::tR(ccu)J/ho::Hyg lys2::NatMX/ LYS2 SYN10/WT10 |
| yYW185 | yMS521, SCRaMbLEd, caffeine tolerant | MATa/α ura3Δ0/ura3::KanMX ho::tR(ccu)J/ho::Hyg lys2::NatMX/ LYS2 SYN10/WT10 |
| yYW186 | yMS548, SCRaMbLEd, caffeine tolerant | MATa/α ura3Δ0/ura3::KanMX ho::tR(ccu)J/ho::Hyg lys2::NatMX/ LYS2 SYN5/WT5 SYN10/WT10 |
| yYW187 | yMS548, SCRaMbLEd, caffeine tolerant | MATa/α ura3Δ0/ura3::KanMX ho::tR(ccu)J/ho::Hyg lys2::NatMX/ LYS2 SYN5/WT5 SYN10/WT10 |
| yYW195 | yMS548, SCRaMbLEd, caffeine tolerant | MATa/α ura3Δ0/ura3::KanMX ho::tR(ccu)J/ho::Hyg lys2::NatMX/ LYS2 SYN5/WT5 SYN10/WT10 |
| yYW196 | yMS548, SCRaMbLEd, caffeine tolerant | MATa/α ura3Δ0/ura3::KanMX ho::tR(ccu)J/ho::Hyg lys2::NatMX/ LYS2 SYN5/WT5 SYN10/WT10 |
| yMS637 | yMS521, SCRaMbLEd, caffeine tolerant | MATa/α ura3Δ0/ura3::KanMX ho::tR(ccu)J/ho::Hyg lys2::NatMX/ LYS2 SYN10/WT10 |
| yMS638 | yMS521, SCRaMbLEd, caffeine tolerant | MATa/α ura3Δ0/ura3::KanMX ho::tR(ccu)J/ho::Hyg lys2::NatMX/ LYS2 SYN10/WT10 |
| yMS639 | yMS521, SCRaMbLEd, caffeine tolerant | MATa/α ura3Δ0/ura3::KanMX ho::tR(ccu)J/ho::Hyg lys2::NatMX/ LYS2 SYN10/WT10 |
| yMS640 | yMS521, SCRaMbLEd, caffeine tolerant | MATa/α ura3Δ0/ura3::KanMX ho::tR(ccu)J/ho::Hyg lys2::NatMX/ LYS2 SYN10/WT10 |
| yMS674 | yMS521 + episomal POL32 | MATa/α ura3Δ0/ura3::KanMX ho::tR(ccu)J/ho::Hyg lys2::NatMX/ LYS2 SYN10/WT10 |
| yMS675 | yMS521 + episomal POL32 | MATa/α ura3Δ0/ura3::KanMX ho::tR(ccu)J/ho::Hyg lys2::NatMX/ LYS2 SYN10/WT10 |

the frequency by which a deleterious SCRaMbLE event, i.e., deletion of an essential gene, is fatal to the cell in which it occurs. Further, by creating a collection of such heterozygous diploids using both *S. cerevisiae* and *S. paradoxus* strains, we have shown that the SCRaMbLE technique can be successfully applied to a wide variety of hybrid strain backgrounds.

Additionally, heterozygous diploid SCRaMbLE can rapidly generate new phenotypes by environmental selection. The pCLB2-Cre-EBD construct we employ for SCRaMbLE displays less background activity than pSCW11-Cre-EBD used in previous work[5] and allows for robust induction of SCRaMbLE by β-estradiol in a single, 6-h experiment. Combinatorial, logic-gate-controlled SCRaMbLE switches (Jia et al.[18]), as well as light-inducible SCRaMbLE (Hochrein et al.[19]) using a split Cre recombinase provide other approaches to further reduce unintentional recombination events.

Yeasts have been used as domesticated microbes for thousands of years, beginning with their importance in the production of

food and beverage. Due to the broad range of environmental conditions imposed on yeast for industrial applications, the generation of strains with higher tolerance for temperature, pH, etc. for industrial applications is an important goal for biotechnology. We demonstrate the rapid improvement of thermotolerance in a Y12-synX heterozygous diploid as one such example. A prior study using adaptive laboratory evolution to increase thermotolerance of *S. cerevisiae* took over 300 generations[20] to achieve this phenotype. That we are able to generate a significant increase in thermotolerance during a short, relatively weak pulse of SCRaMbLE activity in a strain harboring a single synthetic chromosome bodes well for further optimization of this phenotype using iterative cycles of SCRaMbLE and incorporation of additional synthetic chromosomes.

We additionally show the power of SCRaMbLE in heterozygous diploids for discovery of biological function. Using the CBS5829-synX hybrid strain, we demonstrate improvement in caffeine tolerance that, in two of our SCRaMbLEd strains, can at least partially be attributed to an increase in copy number of *POL32*, a gene previously unlinked to caffeine resistance. We originally hypothesized that our sequencing might reveal an increase in copy number of *TOR1*, the kinase subunit of the TORC1 complex which caffeine has been shown to inhibit in budding yeast[21]. While we did not observe this in our 10 sequenced strains, it is certainly possible that an increase in *TOR1* copy number would need to be accompanied by a stoichiometric increase in additional TORC1 subunits in order to confer caffeine/rapamycin tolerance. However, we were able to recapitulate an increase in caffeine tolerance by addition of an episomal copy of *POL32* to our non-SCRaMbLEd parent CBS5829-synX strain.

With recent work demonstrating that aneuploidy is a transient adaptation to heat stress[22] and increases the variability of *S. cerevisiae* response to environmental perturbations[23], it is perhaps unsurprising to observe that multiple stochastic, unlinked SCRaMbLE events can result in the expression of one particular phenotype. Copy number analysis of multiple strains displaying the same phenotype may help us tease out more previously undiscovered biological rules.

SCRaMbLE of heterozygous diploids does, however, have a few shortcomings left to be addressed. While we have found a number of inversions present in SCRaMbLEd strains with increased tolerance of environmental or chemical stresses (Supplementary Table 3), testing the effect of individual inversions on phenotype is more challenging than evaluating the impact of deletions or duplications. Additionally, the stochasticity of SCRaMbLE can result in major disruptions of genome structure, as evidenced by the large deletions in synX of yYW166 and yYW167. Finally, verification of phenotype is a critical part of the heterozygous diploid SCRaMbLE workflow.

We have used two specific strains as illustrative examples of how SCRaMbLE in heterozygous diploids is able to generate both industrially relevant gains in phenotype and previously unknown biological associations. The immense diversity of yeast strains, both domesticated and wild, as well as the completion of more synthetic chromosomes as part of the Sc2.0 project[6,7,24–26] should allow for a more top-down approach to achieve both of these aims going forward. By rationally selecting the wild-type strain and the synthetic chromosome(s) incorporated in a heterozygous diploid, one may increase the chance to achieve a specific phenotypic outcome; as DNA synthesis costs continue to decrease, incorporating neochromosomes harboring specific sets of genes predicted to be most relevant to a phenotype will become more feasible as well. Remarkably, we are even able to reduce to practice interspecies SCRaMbLE by performing it in a *S.cerevisiae/S.paradoxus* interspecies hybrid. This expands the universe of

yeast types in which SCRaMbLE can be used for strain optimization substantially. Even further expansion of the range of mating partners for synthetic chromosome bearing strains could be achieved using physical methods such as spheroplast fusion[27]. Combining a priori knowledge with the stochasticity imparted by the SCRaMbLE system should enable the acceleration of biological discovery and productive industrial microbe evolution.

## Methods

**Strains and media**. All yeast strains are described in Table 1 and Supplementary Table 1. All synX and synVsynX containing strains are derived from BY4741 (*MATa leu2Δ0 met15Δ0 ura3Δ0 his3Δ1*). All oligonucleotides used in this work are available upon request. β-estradiol and caffeine were purchased from Sigma-Aldrich (St. Louis, MO). Rapamycin was purchased from EMD Millipore (Billerica, MA). Yeast strains were cultured in YPD medium or SC dropout plates supplemented with appropriate amino acids and/or drugs. YPGE media was prepared with 3% glycerol and 3% ethanol as carbon sources. Transformations were done using standard lithium acetate procedures.

To construct strains yYW168 and yYW169, the *NatMX* cassette was PCR amplified from pFA6a-5FLAG-natMX6 (Addgene, Cambridge, MA) using Phusion DNA polymerase (New England Biolabs, Ipswich, MA) and primers (Integrated DNA Technologies, Coralville, IA) oMS028 and oMS029 including homology upstream and downstream of the *LYS2* coding sequence. The resulting amplicon was purified with the DNA Clean and Concentrator-5 kit (Zymo Research, Irvine, CA), transformed into yYW117 and yYW139 and plated on YEPD plates. The resulting transformants were replica plated onto YPD plates containing 0.1 mg/mL clonNAT (Gold biotechnology, St. Louis, MO) to select for Nat[R] colonies.

Haploid MATα strains from the Saccharomyces Genome Resequencing Project (SGRP) were purchased from the National Collection of Yeast Cultures (Norwich, UK). Each heterozygous diploid strain was constructed by mating either yYW168 or yYW169 (both *MATa*) to the appropriate *MATα* SGRP strain. The resulting diploid cells were selected on YPD plates containing 0.1 mg/mL clonNAT and 0.2 mg/mL G418 (Santa Cruz Biotechnology, Dallas, TX).

Cloning was done in Top10 *Escherichia coli* grown in Luria Broth (LB) media. To select strains with drug-resistant genes, carbenicillin (Sigma-Aldrich) was used at a final concentration of 75 μg/mL.

Agar was added to 2% for preparing solid media.

**Plasmids**. pRS416-pSCW11-CreEBD and pRS416-pCLB2-CreEBD are available from Addgene.

The *POL32* coding sequence was PCR amplified from BY4741 genomic DNA with 500 bp upstream and 300 bp downstream sequence using Phusion DNA polymerase (New England Biolabs) and primers oMS147 and oMS148. The resulting amplicon was digested with *Bam*HI and *Eco*RI (New England Biolabs), gel purified using the Zymoclean Gel DNA Recovery Kit (Zymo Research), and cloned into pRS416 to create pRS416-*POL32*.

**Primers and oligos**. oMS028 5′ AACTGCTAATTATAGAGAGATATCACAGA GTTACTCACTAgacatggaggcccagaatac 3′

oMS029 5′ TAATTATTGTACATGGACATATCATACGTAATGCTCAACC tcgacactggatggcggc 3′

oMS147 5′ ggatccGTAATGTGCTAGTGACATGAATACT 3′
oMS148 5′ gaattcCTAAATGGGATGACGCTGATG 3′

**SCRaMbLE of heterozygous diploids**. Heterozygous diploid cells were transformed with pRS416-pCLB2-CreEBD and maintained on SC–Ura plates. Cells were grown overnight in liquid SC–Ura media to saturation. Cultures were diluted to a starting OD$_{600}$ of 0.1 in 50 mL of fresh liquid SC–Ura media. β-estradiol was added to a final concentration of 1 μM and cultures were incubated at 30 °C with shaking at 225 RPM for 6 h. Cultures were spun down at 3000×*g* for 3 min and washed three times with water to wash out β-estradiol and cells were plated onto either solid YPD medium or solid YPD medium containing a selective agent.

**Genomic DNA preparation**. To prepare genomic DNA, we used the Norgen Fungi/Yeast genomic DNA isolation kit (Norgen Biotek, Ontario, Canada) according to the manufacturer's instructions.

**Whole genome sequencing**. Paired-end whole genome sequencing was performed using an Illumina 4000 system with TruSeq library preparation kits. The length of each read was 151 base pairs. Quality control was performed using Trimmomatic 0.33 with the parameters LEADING:3 TRAILING:3 SLIDINGWINDOW:4:15 MINLEN:75. Alignments to a custom-made reference genome were done using bowtie2 (2.2.9) software. Mapped reads were subsequently filtered using samtools software. Variant calling was performed with the Genome Analysis ToolKit.

ARTICLE

**Detection of SCRaMbLE events and coverage**. A custom Ruby and Python pipeline based on the results of Shen et al.[24]. was employed to detect SCRaMbLE events. First, unmapped reads containing loxPsym sites were aggregated, and those with fewer than 20 bp on either side of the loxPsym site were discarded. Reads were then trisected into three parts: the left arm, the loxPsym site, and the right arm. The synthetic chromosome was decomposed into segments, with each segment spanning the base pairs between two loxPsym sites (i.e., segment 1 includes bases from bp 1 up until the first loxPsym site, segment 2 from after the first loxPsym up until the second loxPsym site, etc.). The Smith-Waterman local alignment algorithm was used to map the left arm and right arm to segments on the synthetic chromosome and establish new junctions. Additionally, the average coverage across each segment was calculated by summing the coverage at each position in the segment and dividing by the segment length. For SCRaMbLEd strains, this value was compared to that of the non-SCRaMbLEd parent to determine changes in copy number. Segment copy number and new junctions were used to determine SCRaMbLE events that occurred.

**Code availability**. All custom scripts are available upon request.

**Data availability**. All short read data can be found at the SRA database (SRP136404). All plasmids and sequences are available upon request. All other data are available from the authors upon reasonable request.

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

## Acknowledgements

Michael Shen and Yi Wu contributed equally to this work and are therefore listed in alphabetical order. We thank Adriana Heguy and her staff at the Genome Technology Center for assistance with DNA sequencing. The work in China was funded by the National Natural Science Foundation of China (21750001, 21621004, and 21676192), the Ministry of Science and Technology of China (973 Program, 2014CB745100) and the International S&T Cooperation Program of China (2015DFA00960). The work in the US was supported by NSF grants MCB-1026068, MCB-1158201, MCB-1445545, and MCB-1616111. M.J.S. is supported by T32 GM066704 (Bach) and Medical Scientist Training Program grant T32GM007308.

## Author contributions

M.J.S., Y.W., L.M., Y.Y., and J.D.B. conceived the study and designed all experiments. M.J.S. and Y.W. oversaw all experiments. Y.L., H.X., and H.Z. worked on SCRaMbLE experiments screening strains in environmental and drug conditions. B.L., X.L., W.X., and X.Z. assisted with sequencing analysis and verification of phenotypes. M.J.S., Y.W., and K.Y. oversaw all data analysis. M.J.S. and J.D.B. drafted the manuscript with topical input from all other authors.

## Additional information

**Competing interests:** J.D.B. is a founder and Director of the following: Neochromosome, Inc., the Center of Excellence for Engineering Biology, and CDI Labs, Inc. and serves on the Scientific Advisory Board of the following: Modern Meadow, Inc., Recombinetics, Inc., and Sample6, Inc. J.S.B. is a founder and Director of Neochromosome, Inc. and serves on the Scientific Advisory Board of LAM Therapeutics Inc. The remaining authors declare no competing interests.

