## [Peer Review File · Nature Communications]

Reviewers' comments:

Reviewer #1 (Remarks to the Author):

In this paper, Shen et al. describe the development of the collection of heterozygous diploid strains produced by mating haploid strains bearing one or two synthetic chromosomes to various haploid native strains. The resulting strains were subjected to SCRaMbLE (Synthetic Chromosome Rearrangement and Modification by LoxP-Mediated Evolution) to generate diversity in synthetic chromosomes carrying LoxP sites. This strategy was used to improve the heterozygous diploids for several phenotypes.

The results are of great interest to several groups of researchers, including those interested in molecular evolution, genome structure, and industrial microbiology. Moreover, the text is clear and well-structured.

However, while the approach is extremely interesting & innovative, and has an enormous potential, I also feel that the study is a bit preliminary and misses some crucial analyses. Specifically, in my opinion, the main strength of the study is that it reports a new strategy to shuffle the genomes of wild and industrial yeasts. However, the method has not yet been explored in enough detail to merit publication of the current manuscript, especially given that the SCRAMBLE-method in itself has recently been described in a few Sc2.0 papers (just appeared or in press). So, the main novelty in this paper is really the application of outcrossing with non-Sc2.0 strains.

Major remarks & suggestions:

1. Intuitively, one advantage of using the heterozygous diploids would be that you can identify superior (recessive) alleles of the non-S288c parent by deleting the S288c counterpart on the synthetic chromosome. However, this is not investigated/mentioned.
2. One (very interesting) application of heterozygous diploids is if the authors would have developed (a few independent) intercross lines (by consecutive rounds of sporulating and mating), which would result in highly chimaeric chromosomes where recombination sites are integrated across. This way, the authors could scramble with genes from the non-S288c parent as well.
3. Control experiments with diploid S288c are lacking (mating WT haploid S288c with an S288c with the synthetic chromosomes). Would the authors observe different/better/other improvements compared with the heterozygous diploids?
4. The phenotypic improvements could to be quantified better (fitness in liquid medium?) and statistical analyses are needed. The claimed improvements are not always clear from the spot assays alone, and it is hard to know whether the improvements are reproducible (as only one experiment is shown). For example, supplementary figure 2 does not show the difference in rapamycin sensitivity between the non-scrambled parent strain yMS521 and its scrambled derivatives yYW185 and yMS637. Another example is figure 2a that does not clearly show the decreased fitness of strain yYW166 at 37°C.
5. It is a pity that the authors do not track the causative genes/mutations/rearrangements that drive phenotypic changes in at least some of the scrambled strains. This is especially true for the diploids with two synthetic chromosomes (Lines 201 etc), but also for all other proof-of-concepts that are mentioned in this paper. Confirming at least a few causative changes would really prove the merit of the method.

Interesting, but perhaps less essential suggestions

6. As one of the main advantages of working with non-S288c strains might be the identification of genetic improvements that are not solely applicable for S288c, it would be interesting to test the effect of the causative genetic alterations (if they are identified, see comment above) in an industrial strain, to see whether its industrial potential can be increased.

7. When discussing the interspecies hybrids that were developed in their study, the authors mention that their data shows that scramble can be used for 'a virtually infinite variety' of hybrid strain backgrounds. This claim is a bit strong, since the authors only evaluated interspecies crossings between *Cerevisia* and *Paradoxus* (the most closely related species to *Cerevisia*). More species (at least the *sensu stricto* clade) should be tested.

Minor suggestions and typos

1. In all figures, the scrambled and non-scrambled strains could be indicated more clearly to ease the perception of the information.

2. In Fig. 3, adding a summary table containing all the rearrangements taking place in the scrambled strains yMS637 and yYW185 compared to the unscrambled parent strain yMS521 (analogous to that in Fig. 2c) would be useful.

3. Lines 75-77: reference is missing.

4. Lines 105-106: the reference to the paper showing that pCLB2 provides a pulse of Cre-EBD expression once per cell cycle is missing.

5. Lines 124-127: fig 1c should be substituted with fig 1b. The conclusion drawn from Fig. 1b should be different.

6. Lines 161-163: the reference to the paper showing that TIM17 is an essential gene is needed.

7. Lines 181-199: the results reported in this paragraph are a bit counterintuitive. If I understand correctly, both caffeine and rapamycin are inhibitors of TOR kinase cascade and, thus, one can expect them to act similarly on the strains having duplication of the segment containing POL32 gene, but the action is opposite in fact (see Fig. 3c and supplementary Fig. 2). Am I interpreting this wrongly? If not, please discuss this result in more details.

Reviewer #2 (Remarks to the Author):

Here the Sc2.0 team applies the SCRaMbLE technique for generating random genome rearrangements to intraspecies and interspecies hybrids for the first time. In both cases, they demonstrate its utility by selecting for improved traits (relative to the F1 hybrid). They then determine the nature of the rearrangements with whole genome sequencing. For thermal tolerance, they observed overlapping deletions in two independent SCRaMbLEd strains, but they did not validate this particular case, perhaps because there were numerous other candidates and the region was fairly broad. In one case, it led them to a novel candidate gene (POL32) whose dosage was validated and shown to affect caffeine resistance.

This paper describes an exciting new use of the SCRaMbLE technique, and the experiments are rigorous and well done. The paper is well written, and the conclusions are well supported. It is perhaps worth noting that this technique could conceivably be used to determine the genetic basis of phenotypic differences between species in some cases, but that is not done here.

I only have a couple of minor comments that may improve the paper:

First, I did not see any discussion of gene conversions, either between the heterozygous copies of the *S. cerevisiae* genome in the intraspecies hybrid or in the *S. cerevisiae* x *S. paradoxus* interspecies hybrid. Since this is one of the major genetic mechanisms by which hybrids are thought to evolve naturally, it seems like it would be useful to get a baseline for how many gene conversions one sees relative to SCRaMbLE events for a particular number of generations or protocol. In other words, when is it safe to assume SCRaMbLEing is the dominant factor?

Fig. 2b: the inferred deletions have surprisingly high read counts in some cases. Have they masked their genomes for repeats and/or removed ambiguously mapping reads? My best guess is that something like that is the issue, and they just need to adjust the stringency of their computational pipeline. If the issue is not computational, could there be genetic heterogeneity in the culture being sequenced (e.g. some deletions are still present in half the cells)?

Page 8: "virtually infinite" seems a bit over the top, especially since only four intraspecies and one interspecies hybrid have been tested. Maybe "wide variety"?

Page 8: rephrase, "The value of yeast as a domesticated microbe has been evident for thousands of years..." Though used for thousands of years, it was not "evident" until Pasteur that yeasts were what was doing the fermentation.

Response to Referees

Reviewer #1:

Major remarks & suggestions:

1. Intuitively, one advantage of using the heterozygous diploids would be that you can identify superior (recessive) alleles of the non-S288c parent by deleting the S288c counterpart on the synthetic chromosome. However, this is not investigated/mentioned.

We apologize for a lack of clarity regarding this point. One of the strengths of heterozygous diploid SCRaMbLE is indeed that we can observe the effect of both segmental duplications and deletions on the synthetic chromosome, which we tried to illustrate with one example each. The new work included in the revision demonstrating increased thermotolerance as a result of a multi-segment deletion in SCRaMbLEd Y12-*synX* strains is an example of an S288c synthetic chromosome allele deletion that drives a phenotypic observation. We more completely address this in our response to remark 5.

2. One (very interesting) application of heterozygous diploids is if the authors would have developed (a few independent) intercross lines (by consecutive rounds of sporulating and mating), which would result in highly chimaeric chromosomes where recombination sites are integrated across. This way, the authors could scramble with genes from the non-S288c parent as well.

This approach is one we plan to follow up on for subsequent work, but is out of the scope of this particular investigation.

3. Control experiments with diploid S288c are lacking (mating WT haploid S288c with an S288c with the synthetic chromosomes). Would the authors observe different/better/other improvements compared with the heterozygous diploids?

We have performed SCRaMbLE experiments on a heterozygous diploid strain with WT haploid S288c mated with BY4741 containing *synX* and performed selection on both caffeine and high temperature. Thus far we have not observed either thermotolerant or caffeine tolerant colonies arising from this set of SCRaMbLE experiments (**line 228** in the main text and **supplementary figure 5**).

4. The phenotypic improvements could to be quantified better (fitness in liquid medium?) and statistical analyses are needed. The claimed improvements are not always clear from the spot assays alone, and it is hard to know whether the improvements are reproducible (as only one experiment is shown). For example, supplementary figure 2 does not show the difference in rapamycin sensitivity between the non-scrambled parent strain yMS521 and its scrambled derivatives yYW185 and yMS637. Another example is figure 2a that does not clearly show the decreased fitness of strain yYW166 at 37°C.

We have quantified the caffeine tolerant phenotype using plate-reader driven assays in liquid media, as shown in **line 210** in the main text and **revised figure 3c**.

5. It is a pity that the authors do not track the causative genes/mutations/rearrangements that drive phenotypic changes in at least some of the scrambled strains. This is especially true for the diploids with two synthetic chromosomes (Lines 201 etc), but also for all other proof-of-concepts that are mentioned in this paper. Confirming at least a few causative changes would really prove the merit of the method.

We have further interrogated our heat tolerant Y12-*synX* strains to more definitively establish the causative rearrangement for this phenotype, which we are able to recapitulate solely with a deletion of the YJL154C-YJL140W segment identified in figure 2 (**line 168** in main text, **supplementary figure 2**). Additionally, we wanted to be more certain that mutations in the SCRaMbLEd, caffeine-tolerant CBS5829-*synX* genomes were not the drivers of the observed phenotype, which we address from **line 212** in the main text and in **supplementary table S2**.

6. As one of the main advantages of working with non-S288c strains might be the identification of genetic improvements that are not solely applicable for S288c, it would be interesting to test the effect of the causative genetic alterations (if they are identified, see comment above) in an industrial strain, to see whether its industrial potential can be increased.

While we have envisioned using SCRaMbLE in heterozygous diploids to enhance industrially useful phenotypes (tolerance to environmental conditions, etc.), we have not yet used industrial strains to test the resulting rearrangements. The aneuploidy commonly observed in these strains would likely complicate analysis. One future direction we are considering is using SCRaMbLE-in to introduce heterologous pathways encoding production of industrially relevant molecules into our heterozygous diploid collection.

7. When discussing the interspecies hybrids that were developed in their study, the authors mention that their data shows that scramble can be used for 'a virtually infinite variety' of hybrid strain backgrounds. This claim is a bit strong, since the authors only evaluated interspecies crossings between *S. cerevisiae* and *S. paradoxus* (the most closely related species to *S. cerevisiae*). More species (at least the sensu stricto clade) should be tested.

We have successfully performed interspecies crossings between *S. cerevisiae* and *S. eubayanus* as well, but have not fully interrogated their performance in the SCRaMbLE system yet.

Minor suggestions and typos

1. In all figures, the scrambled and non-scrambled strains could be indicated more clearly to ease the perception of the information.

We have updated the color scheme to be more consistent in **figures 2 and 3**.

2. In Fig. 3, adding a summary table containing all the rearrangements taking place in the scrambled strains yMS637 and yYW185 compared to the unscrambled parent strain yMS521 (analogous to that in Fig. 2c) would be useful.

The summary table for detected rearrangements is included as **supplementary table S3**

3. Lines 75-77: reference is missing.

Reference has been added.

4. Lines 105-106: the reference to the paper showing that pCLB2 provides a pulse of Cre-EBD expression once per cell cycle is missing.

Reference has been added.

5. Lines 124-127: fig 1c should be substituted with fig 1b. The conclusion drawn from Fig. 1b should be different.

Reference has been added.

6. Lines 161-163: the reference to the paper showing that TIM17 is an essential gene is needed.

Reference has been added.

7. Lines 181-199: the results reported in this paragraph are a bit counterintuitive. If I understand correctly, both caffeine and rapamycin are inhibitors of TOR kinase cascade and, thus, one can expect them to act similarly on the strains having duplication of the segment containing POL32 gene, but the action is opposite in fact (see Fig. 3c and supplementary Fig. 2). Am I interpreting this wrongly? If not, please discuss this result in more details.

For *TOR* this would certainly be the expectation. But *POL32* is not *TOR*. It is possible that the *POL32* duplication confers resistance to caffeine, but not necessarily rapamycin. However, mechanistic studies of this phenotype seem beyond the scope of this paper, which is focused on SCRaMbLE in heterozygous diploids. We discuss this a little further from **line 271**.

Reviewer #2:

First, I did not see any discussion of gene conversions, either between the heterozygous copies of the *S. cerevisiae* genome in the intraspecies hybrid or in the *S. cerevisiae* x *S. paradoxus* interspecies hybrid. Since this is one of the major genetic mechanisms by which hybrids are thought to evolve naturally, it seems like it would be useful to get a baseline for how many gene conversions one sees relative to SCRaMbLE events for a particular number of generations or protocol. In other words, when is it safe to assume SCRaMbLEing is the dominant factor?

Gene conversion could certainly lead to phenotypic changes, as could other types of genomic changes like aneuloidy. In these experiments, we only saw robust colonies arising in our initial selective conditions from cells that had been exposed to estradiol, thus enabling Cre activity. These observations indicate that Cre-mediated SCRaMbLE is the major driver of phenotype improvement in the conditions probed here.

Fig. 2b: the inferred deletions have surprisingly high read counts in some cases. Have they masked their genomes for repeats and/or removed ambiguously mapping reads? My best guess is that something like that is the issue, and they just need to adjust the stringency of their computational pipeline. If the issue is not computational, could there be genetic heterogeneity in the culture being sequenced (e.g. some deletions are still present in half the cells)?

Excellent point! We have applied a more stringent quality filtering of our mapped reads to remove the ambiguously mapping ones. We are now able to clearly document a more pronounced reduction in the read counts in the inferred deletion areas (**revised Fig. 2**).

Page 8: "virtually infinite" seems a bit over the top, especially since only four intraspecies and one interspecies hybrid have been tested. Maybe "wide variety"?

Suggested change in wording has been made.

Page 8: rephrase, "The value of yeast as a domesticated microbe has been evident for thousands of years..." Though used for thousands of years, it was not "evident" until Pasteur that yeasts were what was doing the fermentation.

Suggested change in wording has been made.

Reviewers' comments:

Reviewer #1 (Remarks to the Author):

In the revised version of the paper the authors addressed most of our remarks. Clarity and significance of the presented data considerably benefit from the revisions made.

Unfortunately, we think that remark nr. 4 was not fully taken into account. Namely, in this comment, we've pointed to the missing clarity of the claimed phenotypical improvements. This problem persists, as the increased thermotolerance of yMS423Δ (stated in lanes 171-173) compared to the non-scrambled parent strain yMS423 is not obvious from supplementary Figure 2, and the decreased tolerance towards rapamycin of the scrambled strains with duplicated POL32 (yMS637 and yYW185, stated in lanes 205-207) compared to the non-scrambled parent strain yMS521 is not obvious from supplementary Figure 4. Indicated experiments still need to be quantified better (fitness in liquid medium?) and statistical analysis still needs to be performed. We were pleased to see that this kind of analysis was done for the caffeine experiment (see fig. 3). However, from the fig. 3c and its legend we were not able to figure out to what level of significance "****" refers to and which samples the authors compare.

We also have a few additional remaining minor comments:

1. In response to remark nr. 1 the authors stated that indeed "The new work included in the revision demonstrating increased thermotolerance as a result of a multi-segment deletion in SCRaMbLEd Y12-synX strains is an example of an S288c synthetic chromosome allele deletion that drives a phenotypic observation".

It is a bit discouraging that the authors did not attempt to figure out the exact gene which is responsible for the phenotype.

2. We would highly appreciate, if the authors could comment on the (surprisingly) high toxicity of the pSCW11-Cre-EBD construct on SC-His medium in the presence of β-estradiol compared to the pCLB2-Cre-EBD (see supplementary figure 1).

3. It would be useful to state the percentages in fig 3c (as in fig 3b).

Reviewer #2 (Remarks to the Author):

My comments have been satisfactorily addressed.

Response to Referees

Reviewer #1 (Remarks to the Author):

In the revised version of the paper the authors addressed most of our remarks. Clarity and significance of the presented data considerably benefit from the revisions made.

Unfortunately, we think that remark nr. 4 was not fully taken into account. Namely, in this comment, we've pointed to the missing clarity of the claimed phenotypical improvements. This problem persists, as the increased thermotolerance of yMS423 Δ (stated in lanes 171-173) compared to the non-scrambled parent strain yMS423 is not obvious from supplementary Figure 2, and the decreased tolerance towards rapamycin of the scrambled strains with duplicated POL32 (yMS637 and yYW185, stated in lanes 205-207) compared to the non-scrambled parent strain yMS521 is not obvious from supplementary Figure 4. Indicated experiments still need to be quantified better (fitness in liquid medium?) and statistical analysis still needs to be performed. We were pleased to see that this kind of analysis was done for the caffeine experiment (see fig. 3). However, from the fig. 3c and its legend we were not able to figure out to what level of significance "****" refers to and which samples the authors compare.

We have performed similar experiments to that of figure 3c for temperature and rapamycin tolerance (now shown in new supplementary figures 3 and 6) to quantify the phenotypes described. In both cases, our observations in liquid media were consistent with the phenotypes observed on solid media. We have indicated in figure legends the levels of significance as well as the sample that served as the basis for the multiple comparisons.

We also have a few additional remaining minor comments:

1. In response to remark nr. 1 the authors stated that indeed "The new work included in the revision demonstrating increased thermotolerance as a result of a multi-segment deletion in SCRaMbLEd Y12-synX strains is an example of an S288c synthetic chromosome allele deletion that drives a phenotypic observation".

It is a bit discouraging that the authors did not attempt to figure out the exact gene which is responsible for the phenotype.

Indeed we did attempt to do exactly this. However, the results obtained were exceedingly complex. As is often the case, this is actually a complex case of interactions between multiple genes. Dissecting all the details will take a major effort. While multiple individual knockouts increased thermotolerance, others decreased it. We therefore feel that while it will be interesting to dissect this "polygenic" phenotype, it is beyond the scope of this first publication documenting the value of SCRaMbLE in heterozygous diploids.

2. We would highly appreciate, if the authors could comment on the (surprisingly) high toxicity of the pSCW11-Cre-EBD construct on SC-His medium in the presence of β -estradiol compared to the pCLB2-Cre-EBD (see supplementary figure 1).

This is probably due to the activity of pSCW11 in G1 and the activity being restricted to G2 for CLB2. We modified the text to telegraph this idea.

3. It would be useful to state the percentages in fig 3c (as in fig 3b).

We looked at the figures and thought perhaps this comment refers to figures 1c and 1b and have added in percentages accordingly. Please correct us if we are mistaken though!

REVIEWERS' COMMENTS:

Reviewer #1 (Remarks to the Author):

Shen et al. 2017: Heterozygous diploid and interspecies SCRaMbLEing

In the revised version of the paper the authors addressed all our remarks. We are satisfied with their response and recommend to accept the manuscript for publication.